# Let-7a-5p, miR-100-5p, miR-101-3p, and miR-199a-3p Hyperexpression as Potential Predictive Biomarkers in Early Breast Cancer Patients

**DOI:** 10.3390/jpm11080816

**Published:** 2021-08-20

**Authors:** Paola Fuso, Mariantonietta Di Salvatore, Concetta Santonocito, Donatella Guarino, Chiara Autilio, Antonino Mulè, Damiano Arciuolo, Antonina Rinninella, Flavio Mignone, Matteo Ramundo, Brunella Di Stefano, Armando Orlandi, Ettore Capoluongo, Nicola Nicolotti, Gianluca Franceschini, Alejandro Martin Sanchez, Giampaolo Tortora, Giovanni Scambia, Carlo Barone, Alessandra Cassano

**Affiliations:** 1Department of Woman and Child Health and Public Health, Fondazione Policlinico Universitario Agostino Gemelli IRCCS, Largo A. Gemelli 8, 00168 Rome, Italy; paola.fuso@policlinicogemelli.it (P.F.); antonino.mule@policlinicogemelli.it (A.M.); damiano.arciuolo@policlinicogemelli.it (D.A.); giovanni.scambia@policlinicogemelli.it (G.S.); 2Faculty of Medicine and Surgery, Università Cattolica Del Sacro Cuore, Largo F. Vito 8, 00168 Rome, Italy; concetta.santonocito@policlinicogemelli.it (C.S.); donatella.guarino@policlinicogemelli.it (D.G.); m.ramundo@piafondazionepanico.it (M.R.); brunella.distefano@guest.policlinicogemelli.it (B.D.S.); armando.orlandi@policlinicogemelli.it (A.O.); ettore.capoluongo@unicatt.it (E.C.); nicola.nicolotti@policlinicogemelli.it (N.N.); gianluca.franceschini@policlinicogemelli.it (G.F.); martin.sanchez@policlinicogemelli.it (A.M.S.); giampaolo.tortora@policlinicogemelli.it (G.T.); carlo.barone@unicatt.it (C.B.); alessandra.cassano@policlinicogemelli.it (A.C.); 3Comprehensive Cancer Center, Medical Oncology Unit, Fondazione Policlinico Universitario “A. Gemelli” IRCCS, Largo A. Gemelli 8, 00168 Rome, Italy; 4Laboratory of Clinical Molecular Biology, Department of Biochemistry and Clinical Biochemistry, Fondazione Policlinico Universitario Agostino Gemelli IRCCS, Largo A. Gemelli 8, 00168 Rome, Italy; 5Department of Biochemistry and Molecular Biology, Faculty of Biology and Research Institute, Universidad Complutense, Av. Sèneca, 2, 28040 Madrid, Spain; cautilio@ucm.es; 6Department of Pathologic Anatomy, Fondazione Policlinico Universitario Agostino Gemelli IRCCS, Largo A. Gemelli 8, 00168 Rome, Italy; 7Department of Science and Innovation Technology, University of Piemonte Orientale, V.le Teresa Michel 11, 15121 Alessandria, Italy; 20001886@studentiunipo.it (A.R.); flavio.mignone@unipo.it (F.M.); 8Biotecnologie Avanzate, Università Federico II-CEINGE, Corso Umberto I 40, 80138 Naples, Italy; 9Medical Management, Fondazione Policlinico Universitario Agostino Gemelli IRCCS, Largo A. Gemelli 8, 00168 Rome, Italy; 10Multidisciplinary Breast Center, Dipartimento Scienze della Salute della Donna e del Bambino e di Sanità Pubblica, Fondazione Policlinico Universitario Agostino Gemelli IRCCS, Largo A. Gemelli 8, 00168 Rome, Italy

**Keywords:** subtypes breast cancer, miRNAs, breast cancer treatment, chemotherapy, integrated therapies, next-generation-sequencing, target therapy, precision medicine, personalized medicine

## Abstract

Background: The aim of this study is to identify miRNAs able to predict the outcomes in breast cancer patients after neoadjuvant chemotherapy (NAC). Patients and methods: We retrospectively analyzed 24 patients receiving NAC and not reaching pathologic complete response (pCR). miRNAs were analyzed using an Illumina Next-Generation-Sequencing (NGS) system. Results: Event-free survival (EFS) and overall survival (OS) were significantly higher in patients with up-regulation of let-7a-5p (EFS *p* = 0.006; OS *p* = 0.0001), mirR-100-5p (EFS s *p* = 0.01; OS *p* = 0.03), miR-101-3p (EFS *p* = 0.05; OS *p* = 0.01), and miR-199a-3p (EFS *p* = 0.02; OS *p* = 0.01) in post-NAC samples, independently from breast cancer subtypes. At multivariate analysis, only let-7a-5p was significantly associated with EFS (*p* = 0.009) and OS (*p* = 0.0008). Conclusion: Up-regulation of the above miRNAs could represent biomarkers in breast cancer.

## 1. Introduction

Breast cancer is a heterogeneous disease and many molecular changes occur during the course of the disease; this is the main cause of treatment failure. This characteristic of breast cancer is reflected on the basis of gene expression pattern classification. It falls under five distinct molecular subtypes including luminal A, luminal B, receptor tyrosine-protein kinase erbB-2 (HER2)-enriched, basal-like, and normal-like subtype. Luminal A breast cancer is hormone-receptor positive (estrogen-receptor (ER) and/or progesterone-receptor (PR) positive), HER2-negative, has low levels of the protein Ki-67, and is low-grade. Luminal B breast cancer is hormone-receptor positive (ER and/or PR positive) and either HER2-positive or HER2-negative with high levels of Ki-67. HER2-enriched breast cancer is hormone-receptor negative (ER and PR negative) and HER2-positive. Triple negative breast cancer (TN) is defined as the absence of estrogen receptor, progesterone receptor, and HER2 expression accounting for approximately 15–20% of all breast cancer patients. The majority of TN patients (up to 70%) overlap with the basal-like gene expression subtype.

Tumor evolution is a unique process for each patient and is influenced by intrinsic genetic variability and external factors such as cancer therapy. Neoadjuvant setting is an ideal scenario to understand tumor evolution at a single patient level, because make it possible to identify molecular changes occurring in tumors due to treatment by comparing pre and post-chemotherapy samples [1,2,3].

Finding the patients most likely to benefit from NAC is a crucial need and increasing experimental and clinical studies are centered on identifying the predictors of long-term benefit. Several surrogate endpoints have been examined in the neoadjuvant setting such as the pCR, which has been identified as a primary endpoint in numerous clinical trials despite the controversies on its power of predicting the outcome [3,4].

It is noteworthy that not all patients with residual disease after NAC relapse, and the prognostic impact of pCR varies among breast cancer-intrinsic subtypes, whereas patients with luminal A-like breast cancer show a low pCR rate, their overall prognosis is favorable, and patients with TN breast cancer show a high pCR rate but may have a poorer outcome; moreover, if all intrinsic subtypes are considered, the prognostic information of pCR is reduced [5,6,7,8,9].

Several studies have been performed to discover molecular breast cancer biomarkers in order to predict response to neoadjuvant therapy.

miRNAs are involved in pathway regulation (one miRNA can target many genes and a single gene can be modulated by several miRNAs), and finally, miRNAs show tissue and cell-specific expression profiles, and their role in the pathophysiology of the disease is supported extensively in the literature [10].

Each miRNA can regulate the expression of several genes; thus, each one can simultaneously modulate multiple cellular signaling pathways. Depending on their modulation (amplification/deletion) and on target gene function (tumor suppressor/oncogene), miRNA can play alternatively an oncosuppressor or oncogene function. MiRNAs expression in tumors can be altered due to epigenetic, genetic, and transcriptional alterations [11,12].

Several studies have demonstrated that many miRNAs are aberrantly expressed in breast cancer, according to breast cancer molecular subtypes and thus potentially play a role of biomarkers for cancer diagnosis and for response to therapy [13].

We hypothesized that miRNA are differently expressed at different steps of the disease, and it could be possible to identify a set of miRNA associated with disease progression or response to therapy and to attribute to them a predictive and prognostic value.

The aim of the present exploratory study was to identify a set of miRNAs able to predict the prognosis of patients who underwent NAC not achieving pCR.

## 2. Materials and Methods

### 2.1. Patients’ Characteristics and Tumor Specimen Collection

All procedures performed in studies involving human participants were in accordance with the ethical standards of the institutional and/or national research committee and with the 1964 Helsinki declaration and its later amendments or comparable ethical standards. This study has the approval of the Ethics Committee of Fondazione Poli-clinico A. Gemelli IRCCS of Rome (Italy) (N protocol 27736/16), and all patients gave written informed consent. We analyzed our database that contains clinical and pathological data on ≈200 cases that underwent neoadjuvant treatment from July 1997 to April 2014 at Fondazione Policlinico A. Gemelli. Patients had measurable breast tumors. Patients were staged according to the American Joint Committee on Cancer (AJCC) Eighth edition [14]. A TRU-CUT biopsy was obtained from each patient. Classification of intrinsic subtypes was defined according to 16th St. Gallen and ESMO guidelines. Histological type, tumor grade, Ki67, ER, PR, and HER2 status were evaluated in the pre-NAC biopsy and in post-surgical neoplastic specimens. Treatment of HER2-negative breast cancer patients consisted of a combination of anthracyclines, taxanes, and cyclophosphamide, while patients with HER2-positive tumors received taxanes and carboplatin combined with trastuzumab, the latter continued after surgery to complete one year of treatment. Patients with ER and/or PR positive tumors received adjuvant endocrine treatment for at least 5 years. Adjuvant radiotherapy was offered according to the national guidelines [15]. The pCR was defined as the absence of any residual invasive cancer on resected breast specimen and on all sampled ipsilateral lymph nodes (ypT0/is ypN0) [16,17] (Table 1).

From the entire database, we selected twenty-four patients homogeneously distributed according to clinical and pathological characteristics not achieving pCR to which the maximum amount of paraffin-embedding samples of both pre- and post-treatment specimen were available (Table 2 and Table 3). In particular, we analyzed pre- and post-NAC samples of the three main molecular subtypes, respectively HER2-positive luminal, HER2-positive non-luminal, and TN subtypes, respectively. For each subtype, we selected four patients with good prognosis and four with poor prognosis.

### 2.2. Purification of miRNA from Paraffin-Embedding Tissue Sections

Standard formalin-fixation and paraffin-embedding (FFPE) procedures always resulted in significant fragmentation and crosslinking of nucleic acid. For each of the two samples (pre- and post-NAC) for each patient, the starting material for RNA purification was made by up to 4 sections of paraffin-embedding tissue with a thickness of 5 µm combined in one preparation. After microdissection, the total RNA was extracted using miRNeasy FFPE Kit (Qiagen) following the protocol of the manufacturer. The concentration and purity of the total RNA was isolated from tissues and was determined by measuring the absorbance in a spectrophotometer (Nanodrop). The QIAseq miRNA Library Kit (Qiagen) was used for miRNA libraries. In an unbiased rapid reaction, adapters were ligated sequentially to the 3′ and 5′ ends of the miRNAs. Subsequently, universal cDNA synthesis with UMI (Unique Molecular Index) assignment, cDNA cleanup, library amplification, and library cleanup were performed following the manufacturer’s recommendation. The integrity and size distribution of the total RNA from the tissue was confirmed using an automated analysis system (Agilent 2100 Bioan-alyzer). Successively, the miRNA sequencing libraries was sequenced using MiSeq^®^ Il-lumina NGS system: the molarity of each sample (in nM) was calculated using the following equation: (X ng/µL)(106)/(112450) = Y nM. Individual libraries were diluted to 4 nM using nuclease-free water and then combined in equimolar amounts.

### 2.3. MiRNA Discovery

#### 2.3.1. Analysis Procedure

The QIAseq miRNA-NGS data analysis software (Qiagen) was used. The results were confirmed manually by aligning the fastqs with the sequences corresponding to all human miRNAs. The miRNA sequences were extracted from the miRBase database [18].

The miRNAs were selected based on the number of reads, and those that differed between pre-NAC and post-NAC were taken into consideration.

#### 2.3.2. MiRNA Target Prediction

To know the potential target site, a computational approach was applied for their validation [19]. The miRNA targets were predicted by the instrument MiRDB [20]. This is an online database for miRNA target prediction and functional annotations with a focus on mature miRNAs. It provides a web interface for target prediction generated by an SVM machine learning algorithm. All gene targets were converted by the Human Gene ID Converter tool into their corresponding NCBI entrez gene ID. Some NCBI-gene ID were searched manually on the HUGO Gene Nomenclature Committee (HGNC) database. Perl language scripts have been made to list the NCBI entrez gene ID for each of the miRNAs to be analyzed. For the mapping of the genes, the KEGG Mapper—Search & Color Pathway tool was used. Only the pathways related to the disease were selected and where the mapped genes were more numerous. The pathways related to the disease were selected in consultation with the bibliographic articles in Pubmed-NCBI.

### 2.4. Statistical Analysis

The primary endpoint was event-free survival (EFS). The secondary endpoint was overall survival (OS). EFS was considered as the time from diagnosis to any relevant event (progression of disease that precludes surgery, local or distant recurrence, or death due to any cause) and was censored at the last follow-up visit. OS was estimated as the interval from diagnosis to death from any cause, and it was censored at the last follow-up visit for the patients still alive. The Kaplan–Meier method was applied for survival probabilities estimation. For univariate analysis, we used the Fisher exact test. Variables (IHC-based molecular subtypes, histological type, tumor grade, Ki67% value, tumor size, clinical lymph node status, cTNM stage, surgery) were included in the multivariate analysis if the univariate *p*-value was <0.05. Multivariate analysis was done using the Cox proportional hazard model. A two-sided *p*-value < 0.05 was considered statistically significant. Analyses were performed using SPSS statistical package version 13.0.

## 3. Results

### 3.1. Patients Charachteristics

Within the entire database, we selected 24 early breast cancer patients, who had undergone neoadjuvant chemotherapy at the IRCCS Fondazione Policlinico A. Gemelli, homogeneously stratified according to clinical and pathological characteristics, and not achieving pCR. In particular, we analyzed pre- and post-NAC samples of eight patients for the following subtypes: HER2-positive luminal, HER2-positive non-luminal, and TN subtypes. Median age at time of study entry was 50.2 years (range, 35 to 72 years). Median follow-up was 80 months, median EFS was 40.7 months, and median OS was 63.3 months.

### 3.2. Clinicopathological Variables and Outcome

We analyzed the correlation between IHC-based molecular subtypes (luminal B/HER2-positive, HER2-positive/non-luminal and TN breast cancer), histological type (ductal invasive breast cancer and others), tumor grade, Ki67% value, tumor size, clinical lymph node status, cTNM stage, surgery, and clinical outcome. Variables showing *p*-values < 0.05 in univariate analyses were used for multivariate logistic regression. However, none of the selected variables were statistically significant at univariate analysis.

3.3. miRNAs and Outcome

Thanks to the computational algorithms and bioinformatics database, we identified 27 miRNAs that were significantly hypo- or hyper-expressed in pre- versus post-NAC samples: hsa-let-7a-5p, hsa-let-7f-5p, hsa-miR-100-5p, hsa-miR-101-3p, hsa-miR-103a-3p, hsa-miR-10a-5p, hsa-miR-10b-5p, hsa-miR-125a-5p, hsa-miR-125b-5p, hsa-miR-126-3p, hsa-miR-143-3p, hsa-miR-191-5p, hsa-miR-196a-5p, hsa-miR-199a-3p, hsa-miR-205-5p, hsa-miR-26a-5p, hsa-miR-26b-5p, hsa-miR-29a-3p, hsa-miR-29c-3p, hsa-miR-30a-5p, hsa-miR-30d-5p, hsa-miR-30e-5p, hsa-miR-510-3p, hsa-miR-92a-3p, hsa-miR-93-5p, hsa-miR-99a-5p, hsa-miR-99b-5p. In Scheme 1, we show the modulation of expression of miRNAs for each subtypes. In Table 4 we presented miRNAs predictive target genes.

Up-regulation of let-7a-5p, mirR-100-5p, miR-101-3p, and miR-199a-3p in post-NAC specimens was significantly correlated with better EFS and OS compared to those with normal or lower expression, independent from breast cancer subtypes.

At subgroup analysis, the overexpression of mentioned miRNAs in post-NAC samples was linked with an improvement in EFS and OS only in HER2-positive non-luminal subtypes (Table 5). Furthermore, when we stratified patients according to a sort of miRNA signature (let-7a-5p, mirR-100-5p, miR-101-3p, miR-199a-3p), we found that patients who concurrently overexpress all four miRNAs experimented a significantly better prognosis in terms of EFS and OS (Table 5; Figure 1, Figure 2, Figure 3, Figure 4 and Figure 5). However, at multivariate analysis, EFS (*p* = 0.009) and OS (*p* = 0.0008) showed a statistically association exclusively with up-regulation of let-7a-5p.

## 4. Discussion

Recent suggestions have revealed that the miRNAs can modulate the expression of oncogenes or tumor suppressor genes. Based on this evidence, miRNAs appear as hopeful biomarkers of breast cancer [21].

Bertoli et al. analyzed the role of several miRNAs in breast cancer and showed that some of them could be useful for diagnostic tools (i.e., miR-9, miR-10b, and miR-17-5p); other miRNAs (i.e., miR-148a and miR-335) may have a prognostic role, while still others (i.e., miR-30c, miR-187, and miR-339-5p) may be predictive of treatment response [22].

In our study, we investigated the potential role of miRNAs as predictors of outcome in early breast cancer patients. We found a significantly differential miRNA expression among some breast cancer subtypes in pre-NAC and post-NAC paraffin-embedding tissue: in particular, we found that the up-regulation of let-7a-5p, miR-100-5p, miR-101-3p, and miR199a-3p in post-NAC samples was significantly associated with better prognosis in terms of EFS and OS, but at multivariate analysis, only overexpression of let-7 was correlated with survival.

Although miR100, miR101, and miR199 did not maintain a statistically significant correlation with survival outcome in multivariate analysis, there is a strong biological rationale supporting their role in breast cancer prognosis and, in our opinion, they deserve further studies.

Interestingly, all these miRNAs have shown to be normally down-regulated in breast cancer and have a role in cancer pathogenesis affecting cell cycle, proliferation, and metastasis diffusion.

Let-7 employs its antiproliferative activities and its tumor-suppressor role by controlling key checkpoints of several mitogenic pathways and by suppressing different oncogenes, including HMGA2, RAS, and MYC [23,24]. Let-7 expression levels have a role as a prognostic marker in several cancers, and the loss of its expression is a marker for less differentiated cancers [25,26]. It is newsworthy that HMGA2 and H-RAS oncogenes are targeted by an induced expression of let-7 in breast cancer cells, and in a murine model of breast cancer, exogenous let-7 delivery represses mammosphere formation, cell proliferation, and the undifferentiated cell population by downregulating both H-RAS and HMGA2 oncogenes [27]. Barh demonstrated that in silico analysis, apart from repressing HMGA2, RAS, and MYC, let-7 may also target CYP19A1, ESR1, and ESR2, thereby potentially blocking estrogen signaling in ER-positive breast cancers [28]. Moreover, Kim et al. affirmed that let-7a inhibits breast cancer cell migration and invasion through the down-regulation of C-C chemokine receptor type 7 expression (CCR7) [29]. Other authors described a new role of let-7a in regulating energy metabolism in neoplastic cells [30]. To underline the role of Let-7 restoration to prevent tumor progression, our study found that the overexpression of let-7 family members in post-NAC samples is associated with a better prognosis in patients with no pCR. From the therapeutic viewpoint, let-7 is an attractive molecule for preventing tumorigenesis and angiogenesis; thus, it could be a potential therapeutic target in several cancers that lose let-7.

miR-100, miR-99a, and miR-99b belong to the miR-100 family. The miRNA-100 controls several genes playing an important modulatory role. mTOR, PI3K, AKT1, IGF1-R, HS3ST2, HOXA1, RAP1B, and FGFR3 are some of the multiple targets of miR-100. Modulating these important genes, miRNA 100 could block proliferation by promoting cell cycle arrest and apoptosis in tumor cells. Furthermore, recent findings suggest that in breast cancer, the miR-100 may act as a pro-differentiating agent for cancer stem cell modulating Wnt/β-catenin pathway and Polo-like kinase 1 gene. It was found that miR100 overexpression has the capability to inhibit the Wnt pathway. Recent evidence showed that miRNA-100 downregulates Polo-like kinase 1 in basal-like cancer, blocking the maintenance and expansion of breast cancer stem cells (BrCSCs), inducing BrCSC differentiation, thus favoring the transition from undifferentiated tumors into well-differentiated ones [31,32]. Petrelli et al. analyzed 123 early node-negative breast cancer tumor specimens: patients were categorized on the basis of the miR-100 expression status. Patients with low miR-100 levels experienced worst distant metastasis-free survival [32]. According to the literature, the miR-100 family could convert an aggressive tumor into a well differentiated, biologically favorable, phenotype. In support of this potential role, miRNA-100 family members are understudied as targets for differentiation therapy: this therapeutic strategy aims to induce the transformation of aggressive cancer cells into well-differentiated ones, which are more sensitive to therapy [31,32].

miR-101 is known to be involved in many important cancer processes such as inhibition of proliferation, chemoresistance, angiogenesis, invasion, and metastasis [33]. According to this hypothesis, several reports showed that the loss of miR-101 is frequent and is associated to a worse outcome in many types of tumors [34,35,36,37,38,39]. Several studies demonstrated that EZH2, a mammalian histone methyltransferase, is emerging as one of the most important targets of miR-101: loss of miR-101 function induces the overexpression of EZH2, which is related to cancer evolution [40,41]. A meta-analysis showed that the down-regulation of miR-101 expression is correlated with a poor prognosis [13]. Liu at al. revealed that a high expression of miR-101 inhibits TNBC progression and increases chemotherapeutic drug-induced apoptosis in TNBC by directly targeting myeloid cell leukemia 1 (MCL-1) [42]. Other authors demonstrated that miR-101 is hypo-expressed in different breast cancer subtypes and stimulates cellular proliferation and invasiveness by targeting Stathmin1 (Stmn1) [43]. According to these findings, our study showed that higher levels of miR-101-3p were correlated with a better EFS and OS, independently from breast cancer subtypes in patients not achieving pCR. Therefore, it is possible to say that miR-101 could be a potential therapeutic target and a novel prognostic factor.

The role in breast cancer progression is unclear regarding miR-199a/b-3p. Some studies showed a loss of miR-199a/b-3p expression in aggressive breast cancer [44]; other evidence demonstrated the ability of miR-199a/b-3p to inhibit proliferation, migration, and multi-drug resistance. miR-199a/b-3p seems to be down-expressed in many types of cancer [45,46,47,48,49,50,51,52]. According to Shou-Qing Li et al., PAK4 could be a possible target of miR-199a/b-3p with an oncosuppresive role: in human breast cell lines, ectopic expression of miR-199a/b-3p blocks the PAK4/MEK/ERK pathway to inhibit breast cancer progression by inducing G1 phase arrest [52]. Xuelong et al. have shown that the hyper-expression of miR-199a-3p inhibits mitochondrial transcription factor A (TFAM) expression, enhancing sensitivity to cisplatin in breast cancer cells. Hence, miR-199a/b-3p could represent a good prognostic and predictive biomarker [53]. It was found that the overexpression of miR-199a-3p regulates the activation of the G protein coupled receptor (GPER), which is involved in tumorigenesis, and suppresses cells’ proliferation, invasion, and epithelial–mesenchymal transition in TNBC [54].

Taking into consideration all our findings, our hypothesis is that miRNA patterns of expression could help identify, in the group of patients not achieving pCR, a population with better outcome. Moreover, in our opinion, the present study is interesting because it gives further support to the fundamental role of the miRNAs in cancer biology and their potential application as target cancer therapies. Several studies have been conducted in order to modulate cellular miRNA levels as inhibiting the oncogenic miRNAs and as restoring the tumor-suppressive ones, with encouraging results [55,56,57,58].

Although larger case series are needed, our findings provide a basis for broader, prospective, and multicenter trials to support the potential role of miRNAs as predictive and prognostic biomarkers not only in early but also in advanced disease. We hope that the identified miRNAs will help in comprehensively understanding their pathway mechanism in breast cancer and improve the therapeutic strategies [59].

## 5. Conclusions

miRNAs have changed our understanding of cell pathway modulation and opened fields not only for the development of novel cancer target therapies but even for new diagnostic tools. At present, important topics in cancer research are discovering the underlying pathways involved in miRNA expression and secretion and understanding miRNA modulation in different phases of cancer progression. Large cohort studies are still required to analyze and confirm the diagnostic, prognostic, and therapeutic application of miRNA.

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
