# Peer review of "Let-7a-5p, miR-100-5p, miR-101-3p, and miR-199a-3p Hyperexpression as Potential Predictive Biomarkers in Early Breast Cancer Patients"

_jpm, 2021, doi:10.3390/jpm11080816_

Round 1

Reviewer 1 Report

In the manuscript entitled “miRNAs as potential biomarkers in early breast cancer patients" authors hypothesized that miRNA are differently expressed at different step of the disease and it could be possible to identify set of miRNA associated with disease progression or response to therapy and to attribute them a predictive and prognostic value.

The study is interesting but details are lacking.

  • In section 2.1 (Patients’characteristics and tumor specimen collection) authors indicated 200 cases but in section 3.1 (Patients charachteristics) they indicated 24 patients. I suggest to insert a summary table with the clinical-pathological information of breast cancer patients, to explain how the 24 patients were selected.
  • I suggest summarizing section 3.2 (Clinicopathological variables and outcome) in a table.
  • Table A compares months in terms of EFS and OS, for example 58 vs 28 for EFS / Let-7a-5p in the all populations. what does the comparison refer to?
  • the authors proposed single photos to compare the prognostic impact of distinct miRNAs, in terms of EFS and OS. It would be better to create two-image panels to exemplify the results.

The manuscript considered suitable for publication in Journal of Personalized Medicine after minor revisions.

Author Response

In the manuscript entitled “miRNAs as potential biomarkers in early breast cancer patients" authors hypothesized that miRNA are differently expressed at different step of the disease and it could be possible to identify set of miRNA associated with disease progression or response to therapy and to attribute them a predictive and prognostic value.

The study is interesting but details are lacking.

  • In section 2.1 (Patients’characteristics and tumor specimen collection) authors indicated 200 cases but in section 3.1 (Patients charachteristics) they indicated 24 patients. I suggest to insert a summary table with the clinical-pathological information of breast cancer patients, to explain how the 24 patients were selected.

We added three tables: Table A with clinical-pathological characteristics of all 200 cases .Table B1-2 with clinical pathological features of the 24 patients. In Matherials and Method section 2.1 we added a small explanation of our selection criteria.

  • I suggest summarizing section 3.2 (Clinicopathological variables and outcome) in a table.

We think that create a table for statistically not significant result may be  not useful.

  • Table A compares months in terms of EFS and OS, for example 58 vs 28 for EFS / Let-7a-5p in the all populations. what does the comparison refer to?

Table A that in the new version of the manuscript will be entitled Table E, refers to the correlation between outcome and the  expression levels of each miRNA in pre and post NAD samples. For example, patients with hyper-expression of oncosuppressor Let-7a-5p in post-NAD samples experienced longer EFS (58 months) than patient without Let-7a-5p hyper-expression (28 months).

  • the authors proposed single photos to compare the prognostic impact of distinct miRNAs, in terms of EFS and OS. It would be better to create two-image panels to exemplify the results.

We provided to create two-image panels for each miRNA.

The manuscript considered suitable for publication in Journal of Personalized Medicine after minor revisions.

Reviewer 2 Report

The submitted manuscript focuses on the identification of a miRNA panel that has the potential to serve as a predictive biomarker for the assessment of neoadjuvant chemotherapy (NAC) efficacy in patients with various types of breast cancer. Since there are limited data on the use of miRNAs for the prediction of treatment outcome in cancer patients, the topic of the submitted paper can attract the remarkable interest of the journal audience. The Introduction and Discussion section are well-written and the Methods are technically sound. However, there are some concerns regarding the paper; they are as follows:

Major concern:

Results are poorly represented and illustrated, especially regarding upregulated miRNA expression levels, which should be shown using bar graphs. Additionally, in the Materials and Methods section, the authors described miRNA target gene prediction, however, no Results on the revealed miRNA target genes are presented. The same is for GO and KEGG functional enrichment analysis – no Results are shown. Moreover, a Table containing the patients’ clinicopathological characteristics should be given.

Minor concerns:

  1. The title should contain the over-expressed miRNA name(s) and the word “predictive” since miRNAs were measured to predict patients’ response to neoadjuvant therapy.
  2. The event-free survival (EFS) refers to the possibility of having a particular group of defined events during the definite time period and the percentage of patients who did not experience that event in that time frame, such as 50%, after treatment. Therefore, this should be reflected upon the description of patients’ outcomes.
  3. In the Introduction, it is recommended to shortly describe the differences between breast cancer subtypes, since the authors studied miRNA expression for three groups of BC patients.
  4. All abbreviations should be explained at first mentioning in the text.

Author Response

Major concern:

  • Results are poorly represented and illustrated, especially regarding upregulated miRNA expression levels, which should be shown using bar graphs.

 We provide a table called “Table C” to represent the miRNAs expression modulation in our specimens.

  • Additionally, in the Materials and Methods section, the authors described miRNA target gene prediction, however, no Results on the revealed miRNA target genes are presented. The same is for GO and KEGG functional enrichment analysis – no Results are shown.

 We provide Table D in which are presented the miRNA gene target predicted. We did not perform a validation analysis on gene target expression in our samples, because it was not in the objectives of our study at that time.

  • Moreover, a Table containing the patients’ clinicopathological characteristics should be given.

 In Tables A, B1-2 are summerized the clinicalpathological characteristics of the patients.

Minor concerns:

  1. The title should contain the over-expressed miRNA name(s) and the word “predictive” since miRNAs were measured to predict patients’ response to neoadjuvant therapy.

We provide to modify title according with your suggestion.

  1. The event-free survival (EFS) refers to the possibility of having a particular group of defined events during the definite time period and the percentage of patients who did not experience that event in that time frame, such as 50%, after treatment. Therefore, this should be reflected upon the description of patients’ outcomes.

 In our casistic the percentage of patient who experienced the event in the period of follow up was 72%.

  1. In the Introduction, it is recommended to shortly describe the differences between breast cancer subtypes, since the authors studied miRNA expression for three groups of BC patients.

We provide to insert in the Introduction section a brief description of the different breast cancer subtypes.

  1. All abbreviations should be explained at first mentioning in the text.

We provided to explain all the abbreviation at the first mentioning.

Round 2

Reviewer 2 Report

The authors have addressed all my concerns reported previously regarding the manuscript.